# Comparison of Bacterial Community in the Jejunum, Ileum and Cecum of Suckling Lambs During Different Growth Stages

**DOI:** 10.3390/microorganisms13092024

**Published:** 2025-08-29

**Authors:** Mengrong Su, Chunmei Du, Wenjie Zhang, Jie Liao, Tao Li, Shangquan Gan, Jian Ma

**Affiliations:** 1College of Coastal Agricultural Sciences, Guangdong Ocean University, Zhanjiang 524088, China; 2Key Laboratory of Livestock and Poultry Healthy Breeding Technology in Northwest China, Xinjiang Agricultural Vocational and Technical University, Changji 831100, China

**Keywords:** suckling lambs, intestinal microbiota, 16S rRNA sequencing, age-dependent variation, intestinal segment specificity

## Abstract

Given that suckling lambs with immature rumen development rely on intestinal microbiota for nutrient utilization, investigating the composition and functional characteristics of their intestinal microbiota is therefore of paramount importance. In this study, 16S rRNA gene amplicon sequencing technology was adopted to characterize and analyze the diversity and composition of the jejunum, ileum and cecum bacterial communities of lambs at 0, 7 and 28 days of age, and to predict the functions of the bacterial communities. The α-diversity analysis results revealed that in the jejunum of lambs, the Chao1, PD, Simpson and Shannon indexes differed significantly among the three age groups (*p* < 0.05). In the ileum, Shannon and Simpson indexes of the 0-days-of-age group were slightly lower than those of the 7 (8.84% and 12.66% reductions, respectively) and 28-days-of-age groups (19.34% and 15.85% reductions, respectively) (0.05 < *p* < 0.10). In the cecum, Simpson and Shannon indexes differed significantly (*p* < 0.05) among the three age groups. At the phylum level, Firmicutes (64.68%) and Proteobacteria (21.76%) dominated the bacterial communities across all intestinal segments, with a total of 42 phyla detected. At the genus level, 19 dominant genera were identified in the jejunum. Except for *Bifidobacterium*, which showed no significant age-related variation (*p* > 0.05), the relative abundance of the remaining 18 genera changed significantly with age (*p* < 0.05). In the ileum, compared with the 0-days-of-age group, the *Lactobacillus* abundance was significantly higher in the 7- and 28-days-of-age groups (*p* < 0.05), while the *Escherichia-Shigella*, *Mannheimia* and *Enterobacter* abundances were significantly reduced (*p* < 0.05). In the cecum, the genera, including *Blautia*, *Sellimonas* and *Ruminococcaceae UCG-014*, exhibited significant age-related differences (*p* < 0.05), whereas other genera showed no significant variation (*p* > 0.05). Collectively, the bacterial community α-diversity, compositional structure and specific genus abundance in the jejunum, ileum and cecum of lambs demonstrated pronounced age-dependent variation and intestinal segment specificity patterns. This study provides a foundation for a deeper understanding of the succession patterns of the early digestive tract microbiota in lambs, and is conducive to the development of early nutrition strategies based on precise regulation of the microbiome.

## 1. Introduction

During the early life stages of mammals, the colonization of gastrointestinal microbiota is a dynamic process that plays pivotal roles in nutrient digestion, immune system homeostasis and metabolic regulation, thereby contributing significantly to the maintenance of host physiological functions [1,2]. Furthermore, the symbiotic relationship between intestinal microbiota and their hosts closely interacts with health status, facilitating gastrointestinal development in the host [3]. Carbohydrates serve as the primary energy source for animals; however, a critical limitation arises from the fact that most mammalian digestive enzymes in the intestinal tract cannot ferment or degrade plant lignocellulosic material. This host limitation underscores the critical role of symbiotic gut microbiota. For instance, in ruminants, their specialized digestive systems and symbiotic relationships with microorganisms enable gut microbiota to secrete abundant cellulases. These enzymes break down plant lignocellulosic substances into absorbable microbial proteins and volatile fatty acids (VFAs), thereby providing essential nutrients and energy for the ruminants [4]. Beyond their involvement in nutrient digestion and absorption, microbial communities in the ruminant digestive tract also influence health and reproductive performance [5]. Under normal physiological conditions, these microbiotas maintain a relatively stable dynamic equilibrium. Nevertheless, when animals are exposed to weakened immunity or other adverse environmental factors (such as a high-carbohydrate diet), the balance of the gastrointestinal microbiota will be disrupted, leading to the occurrence of diseases such as diarrhea, rumen acidosis and enteritis [6]. Additionally, the distribution of gastrointestinal microbiota varies among individuals due to differences in growth stages, species, diet, living environments and other contributing factors [7].

For ruminants, the digestive system predominantly operates in a monogastric hindgut fermentation pattern prior to weaning, transitioning to a forestomach fermentation model post-weaning concurrent with rumen development. The mature rumen harbors a highly diverse and dense microbial consortium comprising bacteria, fungi, archaea and other microorganisms, serving as the central organ of the digestive system. Occupying over 50% of the total stomach volume, the rumen plays critical roles in feed storage, rumination and microbial fermentation/digestion of complex plant-based diets such as cellulose [8]. Given that ruminants primarily rely on rumen microbiota to degrade plant lignocellulosic substances, research on their gastrointestinal microbiota has predominantly focused on forestomach communities, often neglecting intestinal microbiota [9]. For instance, Hao et al. investigated the rumen microbial community dynamics during weaning transitions in dairy calves, and found that no significant difference in α-diversity or β-diversity was observed between pre-weaning and post-weaning groups [9]. However, linear discriminant analysis (LDA) demonstrated that the relative abundance of Fibrobacteres was significantly higher in the post-weaning group compared to the pre-weaning group. Additionally, the network node degree of rumen bacterial communities was markedly elevated after post-weaning (16.54 vs. 9.5) [9]. Similarly, Zhang et al. demonstrated that dietary fiber content can modulate major rumen microbial populations (at phylum and genus levels) in sheep [10]. Low-fiber energy feeds elevated the Actinobacteria abundance and reduce the Cyanobacteria and *Ruminococcaceae* populations, whereas high-fiber diets increased VFAs concentrations and decreased Proteobacteria and *Ruminococcaceae* abundances [10]. These findings suggest that dietary fiber content influences metabolic activities of fiber-degrading bacteria, thereby mediating microbial interactions and competitive dynamics [10]. Furthermore, longitudinal monitoring of rumen bacterial colonization in pre-weaned ruminants using 16S rDNA amplicon sequencing revealed dynamic, stage-specific and functionally structured microbial establishment during rumen development [11,12].

The suckling period represents the most rapid growth phase for lambs and serves as a critical window for the establishment of their intestinal microbiota. However, compared to the extensive research on rumen microbiota, studies focusing on the intestinal microbiome of suckling lambs remain relatively limited. Because the rumen of mammalian ruminants is not yet fully developed and functionally similar to that of monogastric animals, some indigestible nutrients enter the intestines for fermentation [13]. Malmuthuge et al. identified that the intestinal lumen microbiota of one-week-old calves comprises bacteria, fungi, archaea, protozoa and viruses, with Firmicutes, Bacteroidetes, Proteobacteria and Actinobacteria detected across all intestinal segments [14]. Concurrently, microbial communities with low biomass and diversity were detected in the intestines of prenatal lambs, mainly composed of Proteobacteria, Firmicutes and actinomycetes. This indicates that microorganisms are already present in the intestines of prenatal lambs, and the colonization of microorganisms in the fetal intestines begins in the uterus [15,16]. Contrarily, other investigations assert that the fetal gut of healthy sheep remains sterile during gestation, with microbial colonization commencing only after amniotic membrane rupture during parturition [17]. This controversy regarding the timing of intestinal microbiota establishment (prenatal vs. postnatal) in lambs necessitates further systematic investigation to resolve existing discrepancies. The first month postpartum represents a critical window for the development of gastrointestinal microenvironments in lambs. During this period, the structural dynamics of intestinal microbiota exhibit high instability, with colonization processes co-regulated by feeding regimens, dietary transitions, host age, environmental exposures and other interacting factors. Beyond the strong constraints imposed by the above factors on microbial colonization, random colonization in the early stage of life may have a lasting impact on the long-term development of animal microbial communities [18]. As the most abundant and phylogenetically diverse microbial group in pre-weaned ruminants, bacteria play pivotal roles in fiber degradation, immune modulation, VFAs production and gastrointestinal morphological development—processes fundamental to nutrient acquisition and energy metabolism in pre-weaned lambs [18]. Despite high-throughput sequencing technology and data analysis methods have significantly deepened our understanding of the functions and diversity of gut microbiota, the temporal colonization patterns of bacterial communities in suckling lambs remain poorly characterized. During production, lambs usually start to consume solid feed after 7 days of age. By 28 days of age (lasting for three weeks), the microbiota in their intestines may enter a relatively stable stage after undergoing the screening pressure of dietary transition. Therefore, this study aims to analyze the bacterial colonization dynamics of the jejunum, ileum and cecum of lambs at three critical developmental time points (0, 7 and 28 days of age), to clarify the evolution pattern of the intestinal microbiota with age and intestinal segments, and to provide a theoretical basis for promoting the healthy growth and improving production efficiency of young ruminants through targeted regulation of the intestinal microbiota.

## 2. Materials and Methods

### 2.1. Animals and Sample Collection

The animal experimental procedures were approved by the Institutional Animal Care and Use Committee of Guangdong Ocean University (Zhanjiang, Guangdong Province, China; Approval Code: SYXK-2023-032) in compliance with the Regulations for the Administration of Laboratory Animals (issued by the State Science and Technology Commission of the People’s Republic of China, 2013).

A total of 24 male neonatal Hu sheep lambs with similar body weights (2.94 ± 0.22 kg) were selected as experimental subjects. At 0, 7 and 28 days of age, 5 lambs were randomly selected from each time point for slaughter and sampling, resulting in a total of 15 lambs used in the trial. Notably, lambs slaughtered on 0 days of age had not received colostrum, while the remaining lambs were bottle-fed colostrum in three equal portions (300 mL initially, followed by 150 mL each in subsequent feedings) within 18 h post-birth. The colostrum used in the experiment was collected via manual mammary gland massage of ewes, immediately sealed, and stored under frozen conditions. Prior to each feeding, colostrum was thawed in a 36 °C water bath and administered quantitatively via bottle. Beyond colostrum intake, all lambs were bottle-fed ewe milk three times daily at 15% of their body weight. Immediately after birth, all lambs were separated from their dams and, from 7 days of age onward, and began to freely feed on alfalfa hay and starter feed. The detailed composition and nutritional profiles are provided in Appendix A. Experimental lambs were individually identified with ear tags and housed collectively in an 8 m × 8 m dedicated pens with straw bedding renewed every 3 days. Throughout the experimental period, lambs had free access to clean drinking water, with both ewe milk and water maintained at temperatures between 32 °C and 36 °C.

Lambs slaughtered on 7 and 28 days of age were subjected to a 12 h fasting period prior to euthanasia, which was performed via captive bolt stunning followed by humane exsanguination. All slaughter procedures adhered to the National Standard Operating Procedures (GB/T 43562-2023) [19]. Postmortem, the abdominal cavity was opened, and the jejunum, ileum and cecum were carefully isolated using nylon ropes to prevent chyme backflow between adjacent intestinal regions. Chyme samples were collected from each intestinal segment, transferred into sterile centrifuge tubes, and immediately flash-frozen in liquid nitrogen. A total of 45 intestinal content samples were stored at −80 °C until total DNA extraction.

### 2.2. DNA Extraction

Total DNA was extracted from thawed (4 °C) intestinal chyme samples (200 mg) using the ZymoBIOMICS DNA Microprep Kit (D4301, Zymo Research, Irvine, CA, USA). This kit employs an optimized lysis buffer system for rapid (30 min) DNA isolation from high-inhibitor matrices while ensuring DNA purity and compatibility with downstream sequencing applications. Extraction procedures were performed in strict accordance with the manufacturer’s protocols. DNA integrity was verified via 0.8% agarose gel electrophoresis, followed by nucleic acid quantification using the Tecan F200 platform with PicoGreen dye-based fluorescence detection. Extracted DNA samples were normalized to 10 ng/μL using sterile ultrapure water and stored at −80 °C pending further molecular analyses.

### 2.3. PCR Amplification

The 16S rRNA V4 region was amplified from all DNA samples (*n* = 45) using conventional PCR with universal primers 515F (5′-GTGYCAGCMGCCGCGGTAA-3′) and 806R (5′-GGACTACHVGGGTWTCTAAT-3′), each incorporating a 12 nt unique barcode sequence [20]. PCR reactions were conducted in 50-μL mixtures containing 1 × PCR buffer, 5 μL of 2 mM dNTP mix, 3 μL of 25 mM MgSO_4_, 1.5 μL of each primer, 1 U of KOD-Plus-Neo DNA polymerase, and 20 ng of template DNA. Amplification was performed on an Applied Biosystems GeneAmp PCR System 9700 (Thermo Scientific, Waltham, MA, USA) under the following conditions: initial denaturation at 94 °C for 1 min, followed by 25–30 cycles of denaturation at 94 °C for 20 s, annealing at 54 °C for 30 s, and extension at 72 °C for 30 s, with a final extension at 72 °C for 5 min and termination at 4 °C. Technical triplicates were performed for each sample, and PCR products from the exponential phase were pooled equimolarly for library construction. Amplicons were mixed with 6 × loading dye, resolved on 2% agarose gels for target fragment verification, and purified using the Zymoclean Gel Recovery Kit (D4008). Purified products were quantified via Qubit 2.0 Fluorometer (Thermo Scientific, Waltham, MA, USA) and pooled at equimolar concentrations for subsequent sequencing analyses.

### 2.4. High-Throughput Sequencing and Sequencing Data Analysis

Library preparation was performed using the NEBNext Ultra II DNA Library Prep Kit for Illumina (NEB #E7645L, New England BioLabs, Ipswich, MA, USA). Sequencing was conducted on an Illumina NovaSeq 6000 platform using the NovaSeq 6000 SP Reagent Kit v1.5 (Illumina, San Diego, CA, USA) with 2 × 250 bp paired-end (PE) sequencing configuration.

Paired-end reads were merged using FLASH (v1.2.11). Using saber, the sequences of each sample were separated from raw reads based on barcodes, and the barcode sequences were truncated. Sequences were quality filtered using QIIME2 (v2020.2) with the following criteria: exclusion of sequences with mean quality scores < 30, removal of sequences < 200 bp in length, and elimination of reads containing ambiguous bases (N > 0). Denoising and chimera removal were conducted via the Deblur algorithm in QIIME2 (v2024.10), generating amplicon sequence variant (ASV) feature tables and representative sequences. Taxonomic classification was performed using a Naive Bayes classifier trained on the SILVA database (v138), which was also used for ASV annotation. Representative sequences were aligned using QIIME2 (v2024.10)’s multiple sequence alignment tool, and phylogenetic trees were constructed using the FastTree plugin. Homogenize each sample and resampling based on the one with the least data volume among the samples.

All statistical analyses were performed using R (v4.0.5). Phylogenetic diversity (PD) was calculated with the Picante package (v1.8.2), while other alpha and beta diversity metrics were computed using the Vegan package (v2.6-4). Bray–Curtis dissimilarity matrices were generated via Vegan’s vegdist function, and principal coordinate analysis (PCoA) was conducted using the ape package (v5.7-1). Analysis of similarities (ANOSIM) and permutational multivariate analysis of variance (PerMANOVA) were performed using Vegan’s anosim and adonis functions, respectively. Each row is normalized through Z-score standardization (z value = [Actual relative abundance of a certain genus in a specific intestinal region—average relative abundance of the genus in the intestine]/standard deviation), generating a heat map. This method is used to screen dominant genera and conduct cluster analysis at the regional level.

### 2.5. PICRUST2 Prediction

Phylogenetic Investigation of Communities by Reconstruction of Unobserved States 2 (PICRUSt2), an advanced evolution of the original PICRUSt framework, is currently the most frequently cited functional prediction tool based on amplicon sequencing. In this study, we leveraged PICRUSt2 to predict molecular functional potentials for each sample using 16S rRNA gene data. This method infers Kyoto Encyclopedia of Genes and Genomes (KEGG) Ortholog (KO) metabolic pathways by incorporating phylogenetic placement of OTUs along with 16S rRNA gene copy number-corrected abundances. Functional predictions were analyzed at three hierarchical levels of the KO pathway ontology. Following z-score standardization, heatmaps were generated to identify dominant pathways at level 3 and perform regional-level clustering analysis of microbial functional profiles.

### 2.6. Statistical Analysis

All data analyses were conducted using SPSS software (version 26 for Windows, SPSS, Chicago, IL, USA). Data were first tested for normality (Shapiro-Wilk test) and homogeneity of variances (Levene’s test). One-way analysis of variance was employed to compare α-diversity indices and bacterial relative abundances among lambs at 0, 7 and 28 days of age. For data failing normality or homogeneity assumptions, non-parametric alternatives (Kruskal–Wallis test) were applied. Results are presented as mean ± standard error (SE). Statistical significance was set at *p* < 0.05, with trends reported for 0.05 < *p* < 0.10.

## 3. Results

### 3.1. Data Acquisition and Analysis

This study analyzed 45 chyme samples collected from the jejunum, ileum and cecum of lambs across three age groups (Days 0, 7 and 28). Through 16S rRNA gene sequencing, we obtained 1,561,553 raw sequences, averaging 34,701 ± 411 sequences per sample (Appendix A). After quality control processing (denoising, filtering, trimming), 1,527,207 effective sequences were retained, averaging 33,938 ± 403 sequences per sample. Based on the 97% nucleotide sequence consistency among reads, the number of operational taxonomic units (OTUs) after splitting and optimization was 18,062, with an average of 401.4 OTUs per sample (Appendix A). Through Venn diagrams for community analysis, it was determined that there are coexisting and specific OTUs in the intestinal region. Comparative analysis of OTU overlap proportions among jejunum, ileum and cecum showed the following hierarchy: ileum (86.84%, 53 shared OTUs) (Figure 1b) > cecum (73.36%, 67 shared OTUs) (Figure 1c) > jejunum (35.91%, 68 shared OTUs) (Figure 1a). Age-specific comparisons revealed 59.89% overlap (81 shared OTUs) across all three intestinal segments in 0-day-old lambs (Figure 1d), increasing to 81.42% (49 shared OTUs) in 7-day-old lambs (Figure 1e) and decreasing to 62.76% (39 shared OTUs) in 28-day-old lambs (Figure 1f).

### 3.2. Analysis of the α-Diversity Index of Intestinal Bacterial

α-diversity analysis revealed significant age-related and regional variations in microbial community complexity. In the jejunum, all four α-diversity indexes (Chao1, PD, Simpson and Shannon) exhibited significant differences among the three age groups (*p* < 0.05). Conversely, in the ileum, there was no significant difference in the Chao1 index and PD index among the three age groups (*p* > 0.05). However, the Simpson index in the 0-day-old group showed slight decreases of 12.66% and 15.85% compared to the 7-day-old and 28-day-old groups (0.05 < *p* < 0.10), with corresponding Shannon index reductions of 8.84% and 19.34% (0.05 < *p* < 0.10). In the cecum, there was no significant difference in the PD index among the three age groups (*p* > 0.05), while the Shannon index showed significant differences among the groups (*p* < 0.05). The Simpson index of the 0-day-old and 28-day-old groups was significantly higher than that of the 7-day-old group (*p* < 0.05). In addition, compared with the 0-day-old and 28-day-old groups, the Chao1 index of the 7-day-old group decreased by 51.43% and 51.04%, respectively (0.05 < *p* < 0.10). Regional comparisons within the same age group revealed that except for the cecal PD index showing an increasing trend compared to the ileum in 7-day-old group (0.05 < *p* < 0.10), all other α-diversity indexes exhibited significant differences between intestinal regions (*p* < 0.05) (Table 1) (Figure 2).

### 3.3. Analysis of Intestinal Bacterial Beta-Diversity Index

In this study, PCoA based on Bray-Curtis dissimilarity matrices was employed to assess bacterial community structure variations in the jejunum, ileum and cecum of lambs across three age groups (Days 0, 7 and 28). The analysis revealed distinct clustering patterns in jejunal microbiota among the three age groups (Figure 3a), as well as clear separation between three intestinal regions in 0-day-old (Figure 3d) and 7-day-old lambs (Figure 3e). Meanwhile, PERMANOVA was further used to determine whether there were significant differences in the distances between the samples. The results showed that the bacterial communities in the ileum (*p* = 0.051) and the ileum and cecum (*p* = 0.075) of the lambs in the 7-day-old group and the 28-day-old group presented a differential trend, while there was no statistically significant difference in the cecum (*p* = 0.24) community between the 0-day-old group and the 7-day-old group. However, the differences in bacterial communities among different intestinal regions within the same age group and among different age groups within the same intestinal region all reached significant levels (*p* < 0.05) (Appendix A).

### 3.4. Distribution of Dominant Intestinal Bacteria Phyla

At the phylum level, 42 bacterial phyla were detected across the jejunum, ileum and cecum of lambs. The number of bacterial phyla detected in the jejunum, ileum and cecum of the 0-day-old group, 7-day-old group and 28-day-old group was 39, 23 and 22, 16, 27 and 17, 29, 17 and 18, respectively. The gut microbiota was dominated by Firmicutes (64.68 ± 4.328%) and Proteobacteria (21.76 ± 3.747%), with Bacteroidetes (5.32 ± 1.278%) and Actinobacteriota (4.42 ± 0.788%) also representing prevalent components (Figure 4). However, the relative abundances differed among intestinal regions. Jejunal communities were characterized by Firmicutes, Proteobacteria, Bacteroidetes and Actinobacteriota dominance, while ileal and cecal communities showed Firmicutes, Proteobacteria, Actinobacteriota and Bacteroidetes as major constituents. Using a threshold of ≥ 1% mean relative abundance in at least one age group within a given intestinal region, we identified 10 major phyla in the jejunum, 4 in the ileum and 7 in the cecum. Core phyla shared across all three regions included Firmicutes, Proteobacteria, Bacteroidetes and Actinobacteriota. The jejunum and cecum additionally shared Planctomycetota and Verrucomicrobiota, while the jejunum uniquely contained Chloroflexi, Acidobacteriota, Spirochaetota and Proteobacteria. The unique bacterial phylum in the cecum includes Tenericutes. In the jejunum, the 10 major bacterial phyla of lambs in the 3-day-old age groups all showed significant differences (*p* < 0.05). In the ileum, the relative abundance of Actinobacteria in the 28-day-old group showed an 18.8-fold increase compared to the 0-day-old group (0.05 < *p* < 0.1). Firmicutes reached peak abundance in the 7-day-old group, exhibiting a 64% increase relative to the 0-day-old group. Conversely, Proteobacteria demonstrated an inverse trend, with abundance in the 0-day-old group being 14.5 times higher than the 7-day-old group and 44.9 times higher than the 28-day-old group (*p* < 0.05). No significant differences were observed in the relative abundances of other bacterial phyla (*p* > 0.05). In the cecum, the relative abundance of Actinomycota in the 28-day-old group was significantly higher than that in the 0-day-old group and the 7-day-old group (*p* < 0.05); Compared with the 0-day-old group, the relative abundance of Proteobacteria in the 28-day-old group showed an increasing trend (0.05 < *p* < 0.1). Meanwhile, compared with the 0-day-old and 28-day-old groups, the Others in the 7-day-old group decreased by 47.22% and 77.91% (0.05 < *p* < 0.1), and the other major phyla showed no significant difference (*p* > 0.05) (Appendix A).

Based on the criterion of ≥1% mean relative abundance in at least one intestinal region within a given age group, 10, 4 and 7 major phyla were identified in 0-day-old, 7-day-old and 28-day-old groups respectively. Notably, the main phyla in the 28-day-old group and the cecal group shifted from Tenericutes dominance to Proteobacteria (Figure 5). In 0-day-old group, Firmicutes abundance in the jejunum showed slight decreases of 54.28% and 50.86% compared to the ileum and cecum, respectively (0.05 < *p* < 0.1). The relative abundance of Proteobacteria showed no significant difference among various intestinal regions (*p* > 0.05), while the relative abundance of other major bacteria was significantly higher in the colon region than in other intestinal regions (*p* < 0.05). In the 7-day group, Firmicutes abundance in the cecum decreased by 26.64% and 34.85% relative to the jejunum and ileum, respectively (0.05 < *p* < 0.1). Bacteroidetes in the jejunum showed slight reductions of 71.93% and 11.1% compared to the ileum and cecum (0.05 < *p* < 0.1). The relative abundances of other phyla showed significant differences among the various intestinal regions (*p* < 0.05). In the 28-day-old group, the relative abundance of Firmicutes in the ileum and cecum regions was significantly higher than that in the jejunum (*p* < 0.05), while the relative abundance of Proteobacteria, Bacteroides and ε -Proteobacteria in the jejunum region was significantly higher than that in other intestinal regions (*p* < 0.05), and there was no significant difference in the relative abundance of other bacteria among the intestinal regions (*p* > 0.05) (Appendix A).

### 3.5. Distribution of Dominant Intestinal Flora Genus

At the genus level, a total of 1062 bacterial genera (including 271 unclassified taxa) across 42 phyla were detected in the three intestinal regions of lambs across all age groups. Among these, 19, 16 and 31 genera met the criterion of ≥2.5% relative abundance in at least one age group within the jejunum, ileum and cecum, respectively (Appendix A). The dominant bacterial genera shared by these three intestines include *Lactobacillus*, *Lachnospiraceae NK3A20 group*, *Escherichia-Shigella, Olsenella*, [*Eubacterium*] *coprostanoligenes group*, [*Ruminococcus*] *gauvreauii group* and *Bifidobacterium*. In addition, the dominant genera shared by the jejunum and ileum include: *Acetitomaculum*; the dominant genera shared by the jejunum and cecum include: Unclassified *Lachnospiraceae*; the dominant bacterial genera shared by the ileum and cecum include: Enterococcus, *Butyricicoccus*, Enterobacter, *Turicibacter* and Clostridium sensu stricto 1. The dominant bacterial genera in the jejunum also include another 10 genera: *Ralstonia*, *Unclassified Muribaculaceae*, *Pseudomonas*, *Roseburia*, *Bacteroides*, *Weissella*, *Alistipes*, *Streptococcus*, *Psychrobacter and Chryseobacterium*; the dominant bacterial genera in the ileum also include three other genera: *Mannheimia*, *Veillonella* and [*Eubacterium*] *nodatum group*; the dominant genera of bacteria in the cecum also include another 18 genera: *Blautia*, *Sellimonas*, *Caproiciproducens*, *Lachnoclostridium*, [*Ruminococcus*] *torques group*, *Ruminococcaceae UCG-005*, *Ruminococcaceae UCG-014*, *GCA-900066225*, *Christensenellaceae R-7 group*, *Anaerotruncus*, *Hydrogenoanaerobacterium*, *Citrobacter*, *Marvinbryantia*, *Unclassified Ruminococcaceae*, *Ruminococcus 2*, *Prevotellaceae NK3B31 group*, *Phascolarctobacterium* and *Akkermansia* (Appendix A) (Figure 6). In the jejunum, except for the relative abundance of Bifidobacterium which showed no significant difference in each age group (*p* > 0.05), the relative abundance of other dominant bacterial genera showed significant differences in each age group (*p* < 0.05). In the ileum, compared with the 0-day-old group, the relative abundance of Enterococcus in the 28-day-old group tended to increase (0.05 < *p* < 0.1), while compared with the 0-day-old group and the 28-day-old group, the relative abundance of *Acetitomaculum* in the 7-day-old group decreased by 50% and 98.85%, respectively (0.05 < *p* < 0.1). The relative abundance of Lactobacillus in the 7-day-old group and the 28-day-old group was significantly higher than that in the 0-day-old group (*p* < 0.05), while the relative abundance of Escherichia-Shigella, *Mannheimia* and Enterobacter in the 0-day-old group was significantly higher than that in other age-old groups (*p* < 0.05). There was no significant difference in the relative abundance of other dominant bacterial genera among different age groups (*p* > 0.05). In the cecum, compared with the 0-day-old group, the relative abundance of *Butyricicoccus*, Anaerotruncus and *Escherichia-Shigella* in the 28-day-old group showed an increasing trend (0.05 < *p* < 0.1), while compared with the 28-day-old group, The relative abuntivities of *Ruminococcaceae UCG-005*, *Unclassified Lachnospiraceae* and *Bifidobacterium* in the 7-day age group tended to increase (0.05 < *p* < 0.1). The relative abundances of *Blautia*, *Sellimonas*, *Ruminococcaceae UCG-014*, *Hydrogenoanaerobacterium*, *Marvinbryantia*, [*Ruminococcus*] *gauvreauii* and *Olsenella* showed significant differences among lambs of different age groups (*p* < 0.05), while the relative abundances of other dominant bacterial genera showed no significant differences among different age groups (*p* > 0.05).

Based on the criterion of ≥2.5% mean relative abundance in at least one intestinal region within a given age group, 20, 17 and 29 dominant genera were identified in 0-day-old, 7-day-old and 28-day-old groups respectively (Appendix A). The common dominant bacterial genera in these three age groups include *Lactobacillus* and [*Eubacterium*] *coprostanoligenes group*; the common dominant bacterial genera in the 0-day-old group and the 7-day-old group include *Escherichia-Shigella*, *Butyricicoccus*, *Sellimonas*, *Enterococcus*, *Caproiciproducens*, *Lachnoclostridium* and *GCA-900066225*; the dominant genera shared by the 0-day-old group and the 28-day-old group include *Unclassified Muribaculaceae*, *Clostridium sensu stricto 1* and *Bacteroides*; the dominant bacterial genera shared by the 7-day-old group and the 28-day-old group include *Lachnospiraceae NK3A20 group*, *Olsenella*, [*Ruminococcus*] *gauvreauii group*, *Roseburia*, *Blautia* and *Ruminococcaceae UCG-014*. The 0-day-old group also includes another 8 dominant genera: *Enterobacter*, *Mannheimia*, *Streptococcus*, *Chryseobacterium*, *[Ruminococcus] torques group*, *Citrobacter*, *Anaerotruncus* and *Hydrogenoanaerobacterium*; the 7-day-old group also includes two other dominant genera: *Acetitomaculum* and *Veillonella*. The 28-day-old group also includes another 18 dominant genera: *Ralstonia*, *Pseudomonas*, *Bifidobacterium*, *Weissella*, *Unclassified Lachnospiraceae*, *Alistipes*, *Ruminococcaceae UCG-005*, *Christensenellaceae R-7 group*, *Acetitomaculum*, *Psychrobacter*, *Turicibacter*, *Ruminococcus 2*, *[Eubacterium] nodatum group*, *Marvinbryantia*, *Phascolarctobacterium*, *Prevotellaceae NK3B31 group*, *Unclassified Ruminococcaceae* and *Akkermansia* (Appendix A) (Figure 7). In the 0-day-old group, Clostridium sensu stricto 1, Enterobacter and Citrobacter showed no significant differences among various intestinal regions (*p* > 0.05), while other dominant bacterial genera showed significant differences among various intestinal regions (*p* < 0.05). Within the 7-day-old group, no significant regional variations were observed for *Enterococcus*, *Butyricicoccus*, *Caproiciproducens*, *Blautia, GCA-900066225*, *Lachnoclostridium*, *Veillonella* and *Ruminococcaceae* across intestinal regions (*p* > 0.05). In contrast, all other dominant genera exhibited significant regional differences in abundance (*p* < 0.05). Among the 28-day-old groups, *Lactobacillus*, *Blautia*, *Weissella*, *Unclassified Lachnospiraceaea*, *Ruminococcaceae UCG-014*, *Roseburia*, *Marvinbryantia*, *Ralstonia*, *Pseudomonas*, *Psychrobacter*, *Unclassified*
*Muribaculaceae*, *Bacteroides* and *Alistipes* showed significant differences among various intestinal regions (*p* < 0.05), while other dominant genera showed no significant differences among various intestinal regions (*p* > 0.05).

### 3.6. PICRUSt2 Prediction of Molecular Functions in Bacterial Communities

Based on the PICRUSt2 software (v2.5.1), the intestinal flora function of lambs was analyzed to obtain the functional prediction information of bacteria in different samples. The sequencing data were compared using the KEGG database. In the first level of predictions, a total of six gene functional categories were identified: Genetic Information Processing (33.64 ± 0.97%), metabolism (15.74 ± 0.09%), Cellular Processes (14.61 ± 0.62%), Human Diseases (12.30 ± 0.17%), Environmental Information Processing (11.86 ± 0.30%) and Organismal Systems (11.85 ± 0.18%) (Appendix A). In 0-day-old group, Genetic Information Processing and Human Disease-related functions showed no significant regional variation (*p* > 0.05). Compared with the jejunum and cecum, the Cellular Processes of the ileum decreased by 1.92% and 14.34%, respectively (0.05 < *p* < 0.1). All other functional categories exhibited significant regional differences (*p* < 0.05). For 7-day-old group compared with the jejunum and ileum, the Genetic Information Processing of the cecum was reduced by 19.13% and 23.18%, respectively (0.05 < *p* < 0.1), with all other functional categories demonstrating significant regional variations (*p* < 0.05). In 28-day-old lambs, metabolism-related functions showed no significant regional differences (*p* > 0.05), while all other functional categories exhibited significant regional disparities (*p* < 0.05).

At the second level, this study identified 46 gene families in the digesta samples, with 7 meeting the criterion of ≥5% relative abundance in at least one intestinal region across age groups (Appendix A). In 0-day-old group, compared with the ileum and cecum, the relative abundance of gene families associated with cell motility in the jejunum decreased by 10.56% and 31.85%, respectively (0.05 < *p* < 0.1), while families related to folding, sorting and degradation, replication and repair and translation showed no significant regional variations (*p* > 0.05). All other functional gene families exhibited significant regional differences (*p* < 0.05). For 7-day-old group, compared with the jejunum and cecum, the relative abundance of cell motility-related gene families of the ileum decreased by 61.70% and 75.34%, respectively (0.05 < *p* < 0.1), with replication and repair and translation families showing no significant regional variations (*p* > 0.05). All other functional gene families demonstrated significant regional disparities (*p* < 0.05). In 28-day-old lambs, all functional gene families exhibited significant regional variations in abundance (*p* < 0.05).

At the third level, 37 KO pathways were selected (mean relative abundance ≥ 1% in at least one region in three days of age) from an initial pool of 391 pathways. The relative abundance of these pathways was evaluated through the Pearson distance matrix, and combined with heat map analysis, it was indicated that the microbial functional characteristics of the intestine showed significant differences in the three age groups. Further analysis revealed that the ileum of each age group formed independent clusters. Except for the jejunum of the 7th age group which did not cluster with the jejunum of other age groups, the intestines of other age groups all clustered into one category (Figure 8).

## 4. Discussion

There are a large number of microbial communities in the gastrointestinal tract of ruminants, including bacteria, Archaea, and anaerobic fungi. These microbial communities coexist and restrict each other with the ruminant body, maintaining the stability of the intestinal microbial environment of ruminants in a long-term dynamic manner, which plays a crucial role in the health and production of ruminants. While numerous studies have investigated the dynamic distribution and colonization patterns of gastrointestinal microbiota in ruminants, research on microbial communities in the juvenile ruminant digestive system has largely been confined to the rumen and fecal microbiota [21,22]. This focus stems from the rumen’s status as the primary fermentative organ and the non-invasive nature of fecal sampling, which has garnered considerable attention from relevant research scholars [23]. However, emerging evidence indicates that rumen and fecal microbiota inadequately represent microbial communities in other intestinal segments [24]. Consequently, the bacterial communities in each segment of the intestinal tract still lack sufficient characteristic descriptions, and little is known about the colonization of the microbiota in the intestines of young ruminants. Therefore, this study aims to gain a deeper understanding of the bacterial composition and colonization in the jejunum, ileum and cecum regions of lambs on 0, 7 and 28 days of age.

The α-diversity refers to the biodiversity within a specific sample or habitat, with common evaluation metrics including the Chao1, PD, Simpson and Shannon diversity indexes, etc. [25] The Chao1 index primarily estimates the total number of species in a community, with particular emphasis on rare species. The PD (Phylogenetic Diversity) index evaluates the evolutionary history or lineage diversity embedded within a community based on phylogenetic relationships among species. The Simpson diversity index focuses on community dominance, measuring the probability that two randomly selected individuals belong to the same species. The Shannon diversity index integrates both species richness and evenness information. A previous study has demonstrated that among gastrointestinal bacteria in goats, the highest alpha diversity is found in the four chambers of the stomach (rumen, reticulum, omasum and abomasum), followed by the duodenum and posterior intestine (cecum, colon and rectum), while the lowest alpha diversity is observed in the jejunum and ileum [26]. Result mentioned earlier indicates significant variations in bacterial richness and diversity as milk and chyme transition between intestinal regions, which is consistent with the present study. This phenomenon may be attributed to differences in intestinal motility, redox potential, physiological functions and pH levels across distinct intestinal regions. Furthermore, bacterial richness and diversity may differ across age groups even within the same intestinal segment. While lamb intestinal microbiota typically exhibit increasing richness and diversity with age, this ecological succession remains fragile and dynamic, influenced by maternal microbial transmission, environmental temperatures, stressors, host developmental stages and dietary transitions. Our results demonstrate distinct α-diversity patterns across intestinal regions and age groups: in the jejunum, the 0-day-old group exhibited the highest bacterial richness and diversity, followed by the 28-day-old group, with the 7-day-old group showing the lowest values; in the ileum, while α-diversity indices showed no significant age-related differences, the 28-day-old group displayed the highest diversity; in the cecum, the 28-day-old group presented the greatest diversity, followed by the 0-day-old group, with the 7-day-old group again showing the lowest values. However, the α diversity indices among different intestinal regions within the same age group showed significant differences. Lv et al. conducted a study on the effects of early supplementation of starter on rumen development and microbial communities [27]. The study found that the rumen microbial diversity of the group fed with milk replacer and starter (Group S) was lower than that of the group fed with milk replacer (Group C). However, supplementing the animals with appetizers before they were weaned was beneficial for promoting the development of rumen microorganisms [27]. In the present study, lambs in the 28-day-old group received solid feed supplementation in addition to milk, resulting in the highest bacterial diversity and richness observed in both the ileum and cecum of 28-day-old lambs. This phenomenon may be attributed to solid feed intake which can stimulate gastrointestinal tract development, alter the physicochemical environment of the digestive tract and provide more diverse and complex fermentation substrates for gut microbiota. Additionally, the highest bacterial richness and diversity observed in the jejunum of day-0 lambs could be explained by the fact that the jejunum serves as the primary site for receiving nutrients and microorganisms from chyme in the middle digestive tract. Newborn lambs are initially exposed to microorganisms derived from the maternal vagina, feces and skin, and subsequently ingest high concentrations of maternal microbiota through colostrum suckling. These microbial populations rapidly colonize the jejunum, forming a transient state of high diversity. Supporting this observation, a related study found that the intestinal microbiota of bottle-fed lambs within 3 days of age is primarily composed of bacteria from the maternal vagina (46%), ambient air (30%), and pen floor (12%) [15]. These bacterial communities rapidly colonized in the jejunum, resulting in instantaneous high diversity. The 7-day-old group, lacking solid feed intake and remaining in an unstable transitional phase, displayed the lowest α-diversity. In addition, PCoA based on the Bray-Curtis difference matrix and PERMANOVA separated different intestinal regions of the same age group and different age groups of the same intestinal region. The results showed that distinct clustering patterns in jejunal microbiota among the three age groups, as well as clear separation between three intestinal regions in 0-day-old and 7-day-old lambs. The results of this study indicate that the intestinal bacterial community structure of lambs shows dynamic changes with the increase in age and different intestinal regions. This change reflects that during the growth and development of lambs, the population structure of the gastrointestinal flora undergoes dynamic changes according to the host’s gastrointestinal location, diet, environment and age [28,29].

The distribution of microorganisms in the animal’s intestinal tract shows an uneven feature, which will lead to differences in the quantity and types of microorganisms in different intestinal regions. Across all chyme samples, 1062 bacterial genera spanning 42 phyla were identified. Firmicutes and Proteobacteria dominated all intestinal segments, which were in accordance with findings in cattle [30], yaks [31], sheep [32] and goats [33], but different from microbiome profiles of camels [34]. In the intestine of camels, the Firmicutes is the most abundant phylum, followed by Verrucomicrobia. Although ruminants vary in species, the dominant bacterial phyla in some intestinal regions are similar [35]. The ubiquity of these phyla in ruminants indicates that they play a key role in the intestinal microbiota ecology of ruminants. In addition, these two bacterial phyla account for over 85% of all bacteria, while Bacteroidetes and Actinomycetes also have a certain proportion of distribution. Moreover, the relative abundance of these major phyla varies in the intestinal region of lambs at different ages, and there may be a dynamic balance among them. Consistent with previous reports [36,37], Bacteroidetes and Firmicutes were followed by Actinobacteria, Proteobacteria and Spirochaetes as key contributors to polysaccharide and aromatic compound degradation. In addition, it was found in this study that the relative abundance of Firmicutes in the ileum of lambs at 7 days and 28 days of age was significantly higher than that of lambs at 0 days of age, while the Proteobacteria showed the opposite trend. This indicates that in 0-day-old lambs, due to the higher oxygen content in the intestinal tract, it is conducive to the colonization of aerobic/facultative anaerobic bacteria such as Proteobacteria. However, after 7 days of age, a strict anaerobic environment is formed in the intestinal tract, and the relative abundance of Firmicutes increases, which is conducive to the degradation of complex polysaccharides by fiber-degrading bacteria. Zhang et al. employed high-throughput sequencing technology to study the structural composition and spatial distribution characteristics of the intestinal microbial community in small-tailed Han sheep [38]. They found that the dominant bacterial phyla in the cecum and rectum of small-tailed Han sheep were Firmicutes and Bacteroides, while the relative abundance of Firmicutes and Cyanobacteria was higher in the jejunum [37]. Meanwhile, the intestinal microbiota of calves before weaning is mainly composed of Firmicutes, Bacteroides and Proteobacteria. This characteristic of the microbiota in the gastrointestinal tract is similar to that of adult cattle, indicating that the core bacterial community in the intestines of these mature animals was formed in the early stage of life. Our research results are consistent with this. The results of this study indicate that Firmicutes and Bacteroidetes dominate the gastrointestinal microbiota and play a significant role in the digestion and absorption of proteins and carbohydrates in the gastrointestinal tract of ruminants. In addition, the Firmicutes phylum occupies a relatively high proportion in multiple age groups and sites. It not only plays a significant role in the process of energy absorption but also participates in the degradation of oligosaccharides, cellulose and starch, which helps lambs maintain intestinal energy supply, internal environment stability and digest and absorb nutrients. The Proteobacteria phylum may be related to the early immune response in the lamb’s intestinal tract and its resistance to the invasion of external pathogens. This study is conducive to a deeper understanding of the succession patterns of the intestinal flora in lambs during their growth and development, providing a theoretical basis for improving the health and production performance of lambs by regulating the intestinal flora.

In this study, the core bacterial genera, in addition to the above four major phyla, also include the *Verrucomycota* phylum. Relevant studies have shown that some bacterial species, including *Prevotella*, *Bacteroides* and *Ruminococcus*, already exist in the gastrointestinal tract of newborn calves [39,40]. These bacteria mainly participate in the degradation of fiber and starch, and this phenomenon is usually found in adult ruminants. Notably, these microbial communities establish in 1-week-old calves receiving only liquid diets, indicating early-life colonization of beneficial digestive bacteria across gastrointestinal segments even in the absence of solid feed substrates [40]. Our lamb study parallels these findings, demonstrating similar early microbial establishment. Furthermore, Bi et al. identified *Lactobacillus* as a dominant maternal-derived genus in lamb jejunum [15], a pattern corroborated by our detection of consistently high *Lactobacillus* abundance across all examined ages (0, 7 and 28-day-old lambs). As key drivers of small intestinal microbiome succession, Lactobacillus produce lactic acid, creating an acidic milieu that suppresses pathogens while promoting commensal proliferation, thereby maintaining intestinal homeostasis. In addition, the research found that different feeding methods of lambs also have a certain impact on the microorganisms in their intestines. Compared with breastfeeding, bottle feeding significantly increased the abundance of *Escherichia/Shigella*, *Butyricicoccus* and *Clostridium XIVa*. The increase in the content of *Escherichia/Shigella* indicates that artificial feeding may increase the number of potential pathogens [15]. This phenomenon is consistent with the results of this study. With the increase in age, the number of *Escherichia/Shigella* in the jejunum, ileum and cecum of lambs decreases, indicating that the intestinal environment of lambs is gradually maturing and their immune system is in the process of continuous development and improvement. The reduction in *Escherichia*/*Shigella* may reflect the effective control of these pathogens by the host immune system. Different types of bacterial genera exist in different parts of the gastrointestinal tract of ruminants, and the relative abundance of these bacterial genera varies significantly in different parts of the gastrointestinal tract. In 3-day-old lambs, the jejunum is the main site for food digestion and absorption. Research has found that the most abundant genus of bacteria in the jejunum during this period is *Bacteroides* [15], which is different from the results of this study. Another study has found that compared with Bacteroides strains, the main bacterial genera of the Firmiculata phylum are more concentrated in the intestine rather than the rumen [41]. The results of this study indicate that *Lactobacillus* and *Bifidobacterium*, among others, have already begun to colonize initially from the environment or the mother’s body at birth in lambs. They may serve as the foundation for the establishment of beneficial bacteria in the intestines of newborn lambs, facilitating their initial adaptation to the external environment and digestion of breast milk. With the increase in age, the abundance of related genera such as *Ruminococcus* and *Helicobacter* increases. These genera play an important role in the generation of broken-chain fatty acids and the degradation of carbohydrates.

Based on PICRUSt2, the molecular functions of the intestinal bacterial community were predicted. At the KEGG level 2, the most significant functional categories were gene families related to cell motility, aging, translation, replication and repair, drug resistance, antineoplastic and nucleotide metabolism, and folding, sorting, and degradation. This is different from most current studies on weaned lambs over 60 days old or adult ruminants, where the majority of functional genes belong to carbohydrate metabolism, amino acid metabolism, membrane transport and energy metabolism, etc. In this study, although genes related to carbohydrate metabolism and amino acid metabolism also accounted for a certain proportion, they were not the main functional categories. The essence of the results stems from the particularity of the developmental stage of young lambs and the priority of microbial survival strategies. Because lactose in milk can be directly degraded by lactase, microorganisms do not need to be involved in large quantities. It is not until 28-day-old lambs (who have already consumed solid feed) that the need for the degradation of fiber and starch becomes prominent. In the 0-days-of-age lambs, the intestinal microbiota, in order to seize ecological niches and multiply rapidly, thus functional genes such as cell motility, replication and repair and translation are dominant. In the 7 days of age lambs, the intestinal microorganisms adapt to the milk environment. The main functions of the microorganisms are to resist acidic stress and maintain protein stability. Therefore, functional genes such as folding, sorting and degradation and translation are dominant. Among the 28 days of age lambs, due to the intake of solid feed after 7 days of age, the relative abundance of various function-related gene families showed significant differences among different intestinal regions, indicating that the functional development of the microbial community was relatively mature at this time, and the functional differentiation in different regions was obvious. The microorganisms in the intestines of newborn lambs are confronted with three survival pressures: host immune attack (requiring resistance genes), antibacterial substances in breast milk (requiring erroneous protein repair) and niche competition (movement, repair and translation). Therefore, the results of this study reveal the core biological laws of the development of intestinal microorganisms in young ruminants. That is, the transformation from the early stage of planting and survival to the later stage where nutrient metabolism is the main focus. Meanwhile, these findings reveal regional differences in the functional communities of gut microbiota, which are closely related to the characteristics of different gastrointestinal segments and their microbiota interactions [42].

Taken together, this phenomenon is attributed to the complex system of the ruminant digestive tract, where different segments perform varied functions. In 0- to 28-day-old lambs, the diversity and composition of intestinal microbial communities demonstrate significant disparities, likely resulting from physiological and biochemical variations across intestinal regions, including factors such as oxygen content, redox potential, and pH values. The colonization of intestinal microbes progresses from an unstable initial stage to a transitional phase, gradually stabilizing under the influence of multiple factors. In this study, although the lambs were fed a fixed diet after 7 days of age, this period still represents a transitional phase in microbial community development rather than a relatively stable stage. Consequently, the gastrointestinal microbial communities remain dynamic across both temporal and spatial dimensions during this phase. Investigating and elucidating the composition and colonization patterns of gut microbiota in ruminants at different ages and intestinal regions, as well as their interactions, can enhance our understanding of the microbiome’s role in nutrient digestion.

## 5. Conclusions

The results of this study demonstrated that microbial diversity and richness in the jejunum, ileum and cecum were highest on Day 0 and Day 28, with the lowest values observed on Day 7. Firmicutes and Proteobacteria were identified as the core phyla across all intestinal segments, playing pivotal roles in the digestion and absorption of proteins and carbohydrates in the gastrointestinal tract of lambs. Among all detected genera, the *Lactobacillus* exhibited the highest average relative abundance in the jejunum, ileum and cecum, particularly in the ileum. Across the three age groups, except for the Day 0 group where *Escherichia-Shigella* showed the highest relative abundance, *Lactobacillus* dominated over other genera in the Day 7 and Day 28 groups. This study enhances our understanding of the dynamic colonization patterns of intestinal bacterial communities in suckling lambs. Given that Day 7 lambs are in a vulnerable phase of microbial colonization, targeted supplementation with probiotic formulations can be implemented to develop precise nutrition strategies tailored to specific intestinal segments and developmental stages. Furthermore, additional investigations into the precise functions and mechanisms of ruminant gut microbiota are warranted to inform interventions aimed at improving intestinal homeostasis, enhancing production efficiency and optimizing nutritional management in suckling ruminants.

## Figures and Tables

**Figure 1 microorganisms-13-02024-f001:**
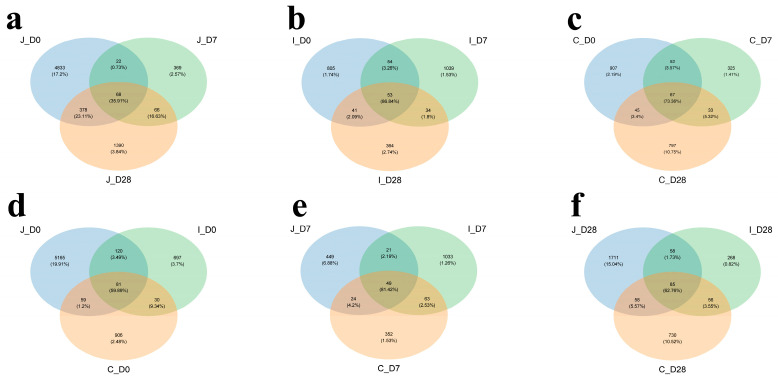
Venn diagram shows distribution of co-present and endemic OTUs in three intestinal regions of lambs at three age groups at 97% similarity level. Common and unique distribution of OTUs in jejunum (**a**), ileum (**b**) and cecum (**c**) of lambs at three age groups. Common and unique OTUs distribution in three intestinal regions of 0-day-old group (**d**), 7-day-old group (**e**) and 28-day-old group (**f**) of lambs.

**Figure 2 microorganisms-13-02024-f002:**
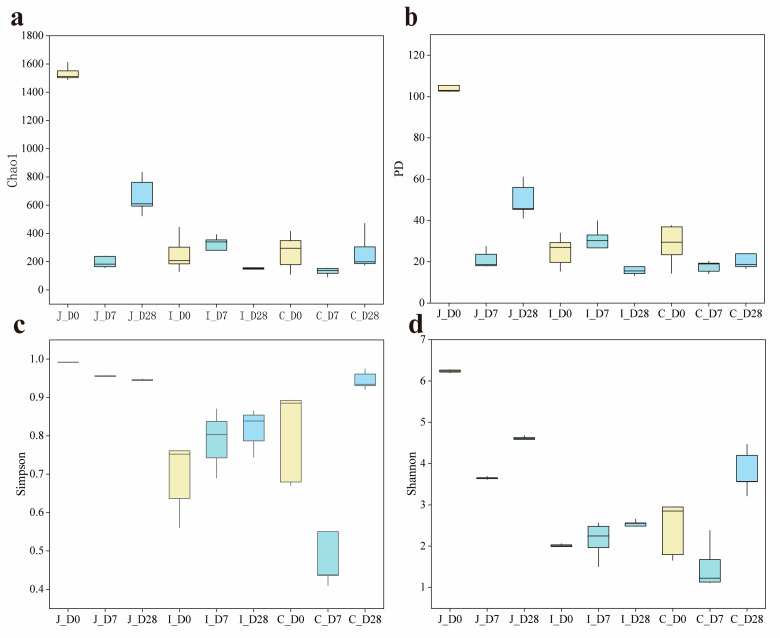
Comparison of α diversity index of three intestinal contents of lambs on 0 d, 7 d, and 28 d (*n* = 5). (**a**) Chao1 diversity index of three intestinal regions. (**b**) PD index of three intestinal regions. (**c**) Simpson diversity index of three intestinal regions. (**d**) Shannon diversity index of three intestinal regions.

**Figure 3 microorganisms-13-02024-f003:**
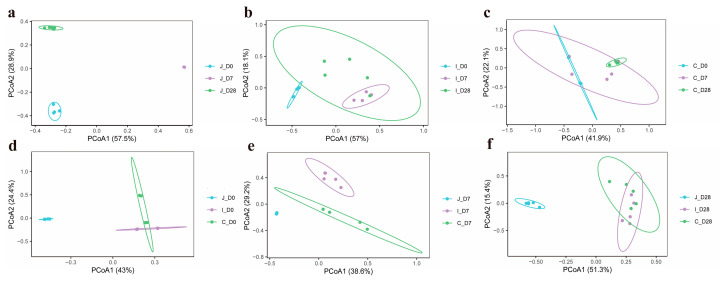
PCoA of bacterial communities in three intestinal regions of lambs on 0, 7, and 28 days of age (*n* = 5). PCoA of bacterial communities in jejunum (**a**), ileum (**b**) and cecum (**c**) region of lambs in 3-day-old age groups. PCoA of bacterial communities in three intestinal regions of lambs in 0-day-old group (**d**), 7-day-old group (**e**) and 28-day-old group (**f**).

**Figure 4 microorganisms-13-02024-f004:**
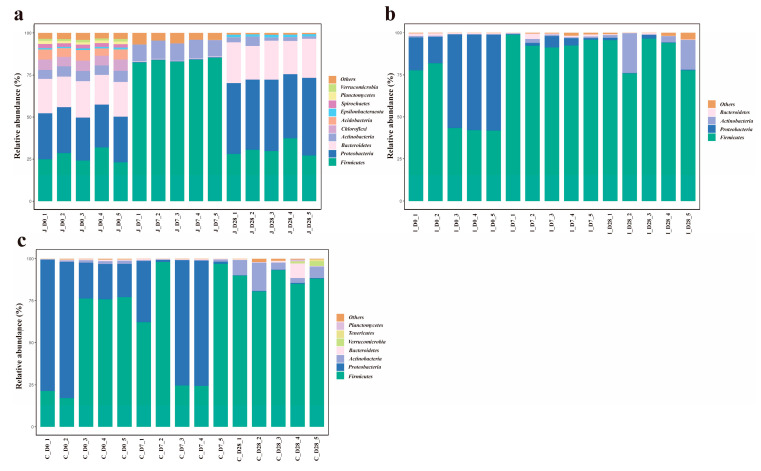
The average abundance of dominant phyla in different age groups of the three intestinal regions (the standard of average relative abundance ≥ 1%). The average abundance of dominant bacteria in the jejunum (**a**), ileum (**b**) and cecum (**c**) of three different age groups.

**Figure 5 microorganisms-13-02024-f005:**
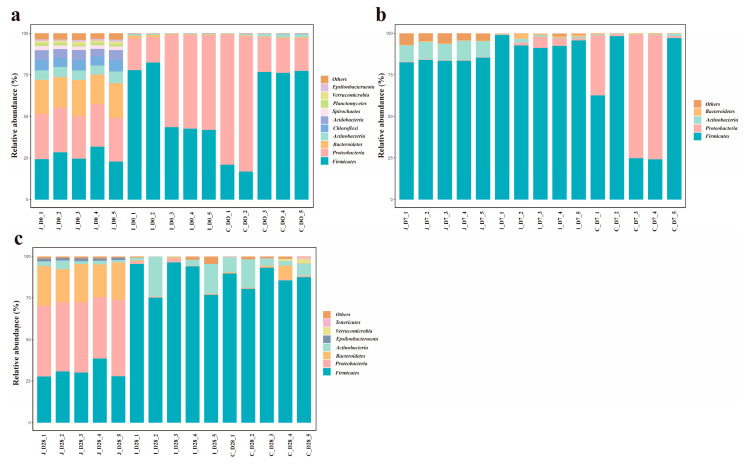
The average abundance of dominant phyla in different intestinal regions within the same age range (the standard of average relative abundance ≥ 1%). The relative abundance of dominant bacterial phyla in each intestinal region of the 0-day-old group (**a**), 7-day-old group (**b**) and 28-day-old group (**c**).

**Figure 6 microorganisms-13-02024-f006:**
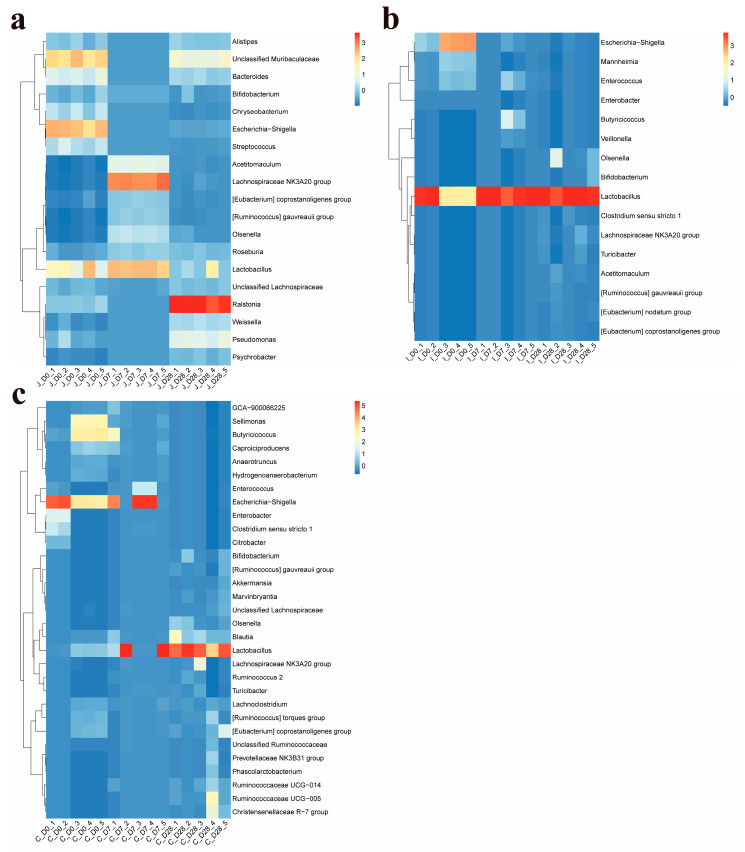
Heat maps of the relative abundance of dominant bacterial genus in the jejunum (**a**), ileum (**b**), and cecum (**c**) of lambs for three age groups (the average relative abundance in at least one area is shown to be ≥2.5).

**Figure 7 microorganisms-13-02024-f007:**
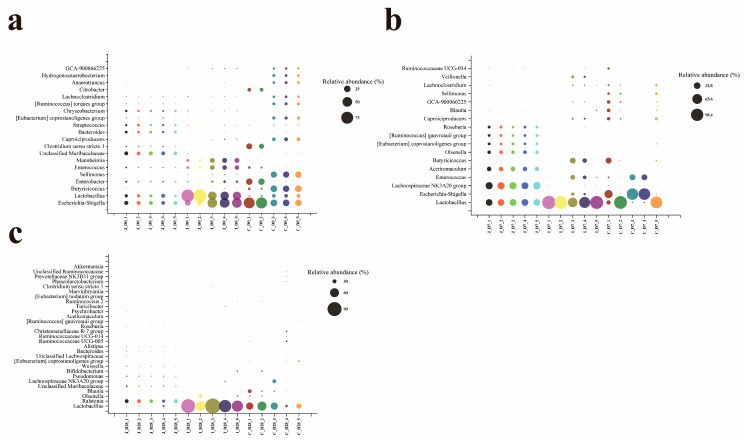
Bubble plots of the relative abundance of dominant bacterial genera in the three intestinal regions of the 0-day-old group (**a**), 7-day-old group (**b**), and 28-day-old group (**c**) (the average relative abundance in at least one region is shown to be ≥2.5).

**Figure 8 microorganisms-13-02024-f008:**
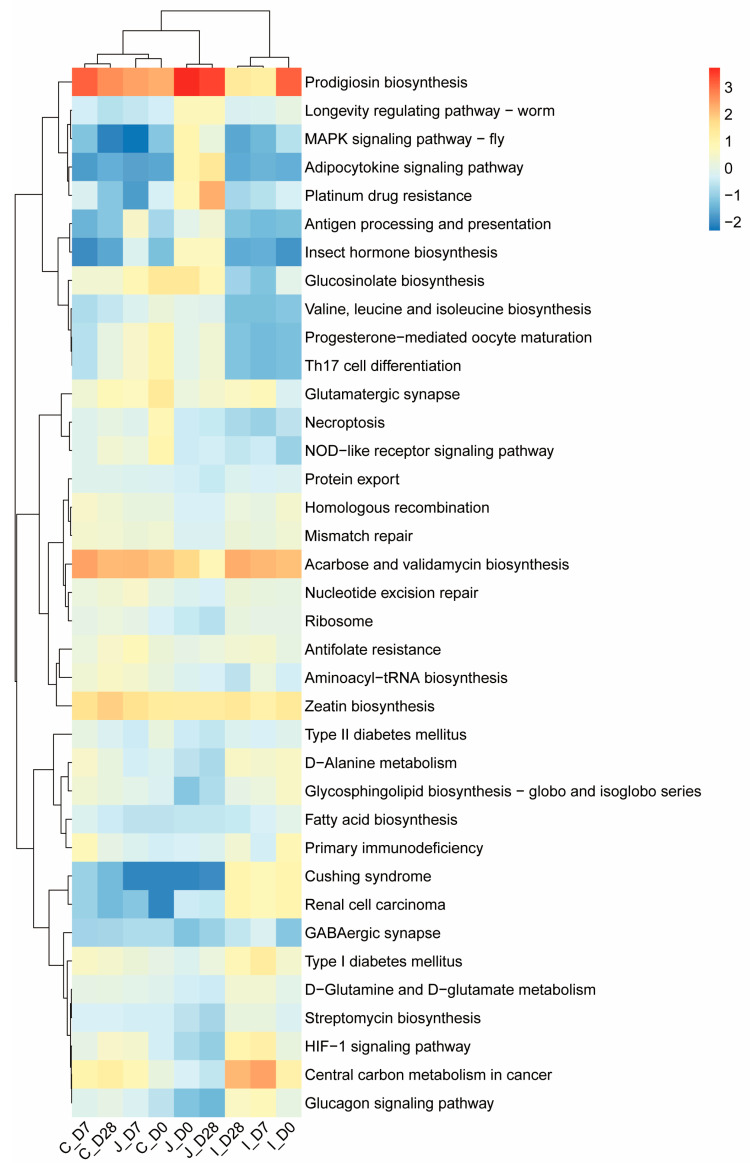
Heat maps of the relative abundance of the third-level KEGG homologous gene functional pathways in three intestinal regions of lambs at three ages (0 d, 7 d and 28 d, *n* = 5) (the average relative abundance in at least one region is shown to be ≥1%).

**Table 1 microorganisms-13-02024-t001:** α -diversity indicators of jejunum, ileum and cecum bacterial communities in three age groups.

Regions	Indexes	0 d	7 d	28 d	*p*-Value
Jejunum	Chao1	1532.98 ± 22.24 ^a^	196.52 ± 17.87 ^c^	664.99 ± 57.43 ^b^	<0.001
	PD	105.23 ± 1.91 ^a^	21.16 ± 1.89 ^c^	49.88 ± 3.73 ^b^	<0.001
	Simpson	0.99 ± 0.0002 ^a^	0.955 ± 0.001 ^b^	0.9438 ± 0.002 ^c^	<0.001
	Shannon	6.24 ± 0.01 ^a^	3.65 ± 0.01 ^c^	4.58 ± 0.05 ^b^	<0.001
Ileum	Chao1	254.19 ± 55.08	289.75 ± 55.09	152.04 ± 8.22	0.124
	PD	25.07 ± 3.34	28.14 ± 4.81	15.67 ± 0.89	0.101
	Simpson	0.69 ± 0.04	0.79 ± 0.03	0.82 ± 0.02	0.051
	Shannon	1.96 ± 0.07	2.15 ± 0.19	2.43 ± 0.14	0.060
Cecum	Chao1	269.95 ± 55.39	131.11 ± 11.85	267.81 ± 55.85	0.088
	PD	28.35 ± 4.35	17.58 ± 1.18	22.67 ± 3.65	0.117
	Simpson	0.80 ± 0.05 ^a^	0.53 ± 0.08 ^b^	0.94 ± 0.01 ^a^	<0.001
	Shannon	2.44 ± 0.29 ^b^	1.51 ± 0.24 ^c^	3.80 ± 0.23 ^a^	<0.001

The values are expressed as mean ± standard error (mean ± SE). The data were derived from the jejunum, ileum and cecum contents samples of lambs on 0, 7, and 28 days of age (*n* = 5). The different superscript letters (a, b, c) in the above values indicate significant differences among the three age groups and the three intestinal sites (*p* < 0.05). Je = jejunum, Il = ileum, Ce = cecum, Chao1 = Chao1 richness index, PD = Phylogenetic diversity index, Simpson = Simpson diversity index, Shannon = Shannon diversity index.

## Data Availability

The sequences from the current study have been deposited in the Sequence Read Archive database of National Center for Biotechnology Information with the accession number PRJNA1293659.

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
