# Peer review of "Comparison of Bacterial Community in the Jejunum, Ileum and Cecum of Suckling Lambs During Different Growth Stages"

_microorganisms, 2025, doi:10.3390/microorganisms13092024_

Round 1
Reviewer 1 Report
Comments and Suggestions for Authors
Dear authors, respectfully, I believe this manuscript addresses a valuable topic and demonstrates potential. However, several issues currently affect the overall quality and clarity of the work. The most significant concern is the use of a generic and overly broad writing style, which diminishes the scientific impact and perceived relevance of the study:
Abstract: My suggestion is to add numbers to improve your description because describing "singinificatively, greater or lesser" is generic. For example, X bacteria showed a 25% higher amount in 28 dyas than in 15 days. It is important that in the abstract being described the highltihs of the study and no generic descriptions.
Lines 16-21: “Show a trend”, “was signifcatively”, etc ….Ok, but what highlith information this description gives? My suggestion is to rewrite or remove this statement and add a more specific description.
Lines 21-23: “Varied” ok, but how was this variation? Avoid generic descriptions.
Lines 26-27: How was this variation? Avoid generic descriptions.
Lines 27-32: This is not a good conclusion for the abstract, The conclusion should be directly and clear.
Keyword: My suggestion is to use keywords other than the title and review the keyword suggestion in the author's instructions.
Introduction: I like this introduction because it has relevant information. My suggestion is to add numbers to improve your description because in the current form it appears a generic description.
Line 42: There is a lack of logical connection between the ideas presented. I recommend rewriting this section to establish a clear and coherent link between concepts, which will enhance the flow and readability of the text.
Lines 55–57: Please provide a reference to support this statement. Claims of this nature should be substantiated by existing literature to ensure scientific credibility.
Lines 65–67: This statement also requires a supporting reference. Adding a citation will strengthen the argument and align the discussion with scholarly standards.
Line 68: The reference number is missing. Ensure that all in-text citations are properly numbered and correspond to the correct source in the reference list.
Line 73: A reference number is also missing here. Please include the appropriate citation to maintain consistency and academic integrity.
Lines 74–77: Add a reference to support the statement made in this section. Referencing relevant literature will provide context and validation for the information presented.
Lines 116–121: Two distinct objectives are described here, but the study appears to have a single overarching objective. Consider rephrasing this section to unify the objectives into one clear, concise statement that accurately reflects the aim of the study.
Results. The current descriptions in the Results section are overly generic and rely heavily on referring to tables and figures without providing sufficient interpretation or detail. To enhance the scientific quality of your writing, consider presenting key findings with specific data values (e.g., percentages, concentrations, or other relevant units) rather than stating that something was simply "higher" or "lower." For example, in line 269, instead of stating that a value was higher, specify by how much - e.g., “X treatment resulted in a 35% increase compared to the control group.” This approach offers greater clarity and strengthens the impact of your findings.
Lines 259-263: “alfa-diversity”, “chao1”, “Simpson”, “Shanon”, etc. These terms may be familiar to experienced researchers, but it is important to consider that new readers regularly enter the scientific community. Therefore, it is recommended to provide brief definitions or explanations of these indices - either in the Introduction or Discussion section - to ensure clarity and accessibility for a broader audience.
Discussion: Place the results in the results topic. The topic of discussion is very speculative. The discussion should focus on explaining how the results were obtained. For this, add theories, hypotheses or statements about how you obtained your results, whether biologically, metabolically, physiologically, environmentally, etc. In the current situation, the discussion is a good general review and results description (Here the results should be avoided so as not to be repetitive); however, you need to make a specific description of how the results were obtained.
You may retain the text currently presented in the Discussion; however, it is important to go beyond simply stating what the results promote or suggest. The discussion should also explore the underlying mechanisms or factors that may have promoted or led to these results. In other words, do not focus solely on the implications of the findings - be sure to also discuss the potential biological, environmental, or experimental conditions that explain why these outcomes occurred. This will provide a more balanced and insightful interpretation of the data.
Line 488: Please describe the mean abundance of the mentioned microbial groups in the rumen. If data are available, also include their relative abundance in different parts of the intestine. Providing these values will offer a more comprehensive understanding of the microbial distribution and contribute to a more informative interpretation.
Lines 521–528: This section presents multiple ideas without supporting references, which weakens the scientific validity of the discussion. Moreover, the descriptions are overly general and speculative. To improve the quality and credibility of this section, the authors should provide specific data from other studies, include appropriate references, and clearly explain how these findings support the hypotheses proposed to explain the results. This will enhance both the clarity and scientific grounding of the discussion
.
Conclusion: The current conclusion is overly broad and lacks precision. A strong scientific conclusion should be concise, direct, and specific. It is essential to begin with the main finding or key highlight of the study, rather than a generic or introductory statement that adds little value for the reader. I recommend rewriting the conclusion to clearly state the primary result, followed by any supporting outcomes or implications. Avoid vague language and focus on delivering a clear takeaway that reflects the core contribution of the research.
Reviewer 2 Report
Comments and Suggestions for Authors
Title
The title of the paper reflects the scope of the research conducted.
Abstract
The abstract is well-written and concludes.
Introduction
The introduction to the research topic and the literature selection are appropriate.
Materials and Methods
Regarding the number of animals used in the experiment, n=15, 5 for each period of 0, 7, and 28 days of life, is too small to obtain reliable, statistically confirmed results. The minimum is n=6.
Results
The results chapter is presented in the form of one table and eight graphs, along with descriptions.
Discussion
The chapter is well-written and should be supplemented with the following information:
Please directly reference the experimental results regarding gut microbiota to the research results of other researchers on this topic.
General Comment
The article submitted for review is interesting from both a scientific and application perspective. It requires minor corrections and additions.
Round 2
Reviewer 1 Report
Comments and Suggestions for Authors
Dear Authors, Reviewers provide suggestions with the primary aim of improving the quality and clarity of the manuscript. However, it is ultimately the responsibility of the authors to decide whether to fully accept, partially accept, or respectfully reject those suggestions.
Thank you for your continued efforts in improving the manuscript.